# A Hybrid Approach to Surface Engineering Based on Laser Texturing and Coating

Matilde Barili [1], Adrian H. A. Lutey [1,*], Corrado Sciancalepore [1] and Luca Romoli [2]

1   Dipartimento di Ingegneria e Architettura, Università degli Studi di Parma, Parco Area delle Scienze, 181/A, 43124 Parma, Italy
2   Dipartimento di Ingegneria Civile e Industriale, Università di Pisa, Largo Lucio Lazzarino, 56122 Pisa, Italy
*   Correspondence: adrian.lutey@unipr.it; Tel.: +39-0521906029

**Abstract:** A hybrid approach based on laser texturing and surface coating for the combined modification of surface topography and chemistry has been proposed to provide a versatile approach for the development of functional surfaces. The experimental procedure comprised nanosecond pulsed laser texturing of AISI 304 stainless steel substrates followed by the deposition of thin (<1 μm) coatings with two different technologies, sol–gel deposition and PE-CVD, with the aim of independently modifying the surface topography and chemical composition. Laser texturing with different scanning strategies achieved a variety of surface morphologies with an arithmetic mean height ($S_a$) in the range 0.2–6.4 μm. Coatings were then deposited on laser-textured substrates to quantify the deposition effectiveness and the influence of the coating type and parameters on the resulting surface topography and chemistry. Sol–gel deposition was found to be more effective with a polymeric interlayer, improving adhesion between the coating and the textured surface; however, this also led to an increase in Sa of approximately 0.5 μm. Conversely, PE-CVD was effective in modifying the surface chemistry while inducing no measurable differences in surface morphology, effectively decoupling the texturing and coating processes. Analysis of the surface chemistry showed a higher concentration of silicon for PE-CVD than sol–gel deposition and therefore a more pronounced effect on the surface chemical composition.

**Keywords:** surface engineering; surface topography; surface chemistry; laser texturing; sol–gel; PE-CVD

## 1. Introduction

Surface engineering has seen a significant uptake over recent years as industry has sought to improve the functionality, efficiency, safety, and lifespan of products while reducing the consumption of energy and raw materials [1]. Generally speaking, surface engineering aims to optimize the behavior and functionality of surfaces and substrates during interaction with the surrounding environment by developing new solutions to improve functional characteristics in terms of physical [2], chemical [3], mechanical [4], optical, and electrical [5] properties, as well as wear resistance [1] and wettability [6]. Due to its versatility, surface engineering has found a wide range of applications in transport, construction, defense, energy, electronics, medicine, food, chemical production, and resource extraction. Despite significant progress, in which laser technology has played a fundamental role [7–9], there is continued interest in developing new treatments and coatings that can achieve higher performance in terms of specific functionalities. One example is the ongoing development of surfaces with tailored wettability in relation to polar and nonpolar liquids, with applications including antibacterial [10] and antifouling surfaces [11], chemical shielding [12], and heat exchangers [13]. Another relevant field in which the synergic effects of surface morphology and chemistry are of the utmost importance is that related to tribology, with applications primarily relating to wear resistance [14], lubrication [15], and the performance of cutting tools [16].

The two most widely applied approaches to surface engineering are surface texturing and coating. The former entails modifying the original surface topography in a controlled manner to produce regular asperities and depressions, while the latter involves altering the chemical composition via the deposition of a layer of material on the substrate. Amongst the various applications of surface engineering, texturing and coating have seen widespread application in the production of hydrophobic surfaces [17,18], which often attempt to mimic naturally occurring surfaces [18–20]. The development of oleophobic surfaces is also of importance owing to rapidly growing interest in the fields of marine antifouling, oil-resistant pipelines [21], adhesive joints, lubrication, sealing [22], heat exchangers, and fuel supply systems [23]. A combination of very low surface energy and adequate surface roughness is fundamental for the fabrication of such surfaces owing to the low surface tension of oil and other organic liquids [24]. The Young, Wenzel, and Cassie–Baxter models provide theoretical bases for the distinct roles of chemical composition and surface topography in functional surfaces with tailored wettability [25,26]. In relation to wear resistance, the synergic effects of surface topography and chemistry derive from the capability of the texture to shield coated regions of the surface from direct contact, protecting them against damage due to mechanical friction [14]. In addition, the flowing speed of water-based lubricants over texture surfaces can be increased via superhydrophobic coatings, providing greater hydrodynamic pressure on the coated surface in severe conditions such as those related to metal cutting [16].

Numerous methods have been tested for the manufacturing of synthetic surfaces with tailored surface topography and/or chemistry, including top-down approaches such as etching, lithography, anodization [27], and laser texturing [28], and bottom-up approaches such as electrodeposition [29], electrospinning [30], hydrothermal methods [31], spray coating [32], sol–gel methods [33], and chemical vapor deposition (CVD) [34]. While some of these approaches modify both the surface topography and the chemistry, these aspects often depend on one another and require a compromise to achieve the best outcome. For example, short and ultrashort pulsed laser technology has shown promise in achieving simultaneous changes to surface topography and chemistry over large areas, leading to hydrophilic or hydrophobic behavior on titanium and steel [35]; however, chemical changes induced by laser processing are largely uncontrolled and depend on factors such as the surrounding atmosphere and temperature during laser exposure, as well as ageing phenomena over subsequent days and weeks [35,36]. Chemical treatment with perfluorooctanoic acid (PFOA) has also been shown to be highly effective at producing omniphobic surfaces [37] but poses major issues in terms of environmental impact, health, and safety. New approaches are therefore required to improve the performance and flexibility of processes for the production of engineered surfaces with independently tailored topography and chemistry.

The present work proposes a hybrid approach based on laser texturing and surface coating for the independent modification of surface topography and chemistry with the aim of providing a versatile approach for the development of new functional surfaces. Nanosecond pulsed laser ablation is combined with sol–gel deposition and plasma-enhanced chemical vapor deposition (PE-CVD). Given the ability to select specific materials for sol–gel deposition and PE-CVD processes, as well as differentiate between the two coating processes based on the industrial requirements at hand, this approach opens up a wide range of possibilities for improved surface engineering as well as the potential to develop higher performance, more robust, and more environmentally compatible surfaces with tailored properties. Laser processing on its own typically leads to precise control over surface topography but poor control over surface chemistry, while sol–gel deposition and PE-CVD allow the modification of surface chemistry but very limited control over surface topography. The hybrid method adopted in the present work therefore provides unique advantages over other approaches to surface engineering owing to the fact that precise modification of surface topography and surface chemistry can be performed independently. The proposed approach therefore represents a feasible pathway to the development of

oleophobic surfaces, which have very specific requirements in terms of surface topography and chemistry that are currently difficult to achieve with existing approaches.

## 2. Materials and Methods

The adopted experimental procedure encompassed nanosecond laser texturing of metallic substrates followed by the deposition of thin (<1 μm) coatings with the aim of independently modifying the surface topography and chemical composition of the surface. Two separate coating technologies were applied, including sol–gel deposition and PE-CVD. A schematic of the proposed hybrid method is presented in Figure 1. Optimized and stable conditions for sol–gel and PE-CVD treatments were chosen based on previous experience and the existing literature. Various laser parameters and scanning strategies were employed to assess the influence of surface topography on the resulting outcome and determine whether sol–gel and PE-CVD treatments, further to modifying surface chemistry, led to appreciable changes in topography.

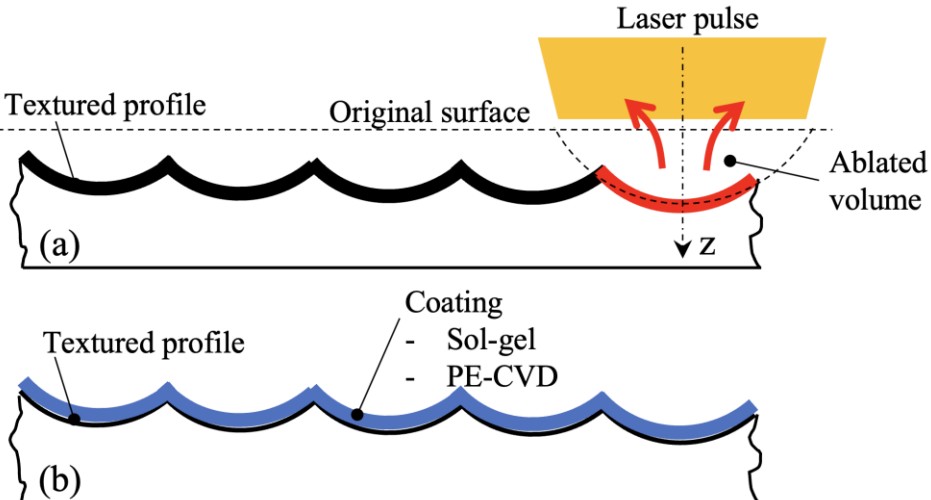

**Figure 1.** Schematic representation of proposed hybrid method: (**a**) Laser texturing; (**b**) subsequent coating treatment.

### 2.1. Substrate

AISI 304 stainless steel samples were employed for all experiments, measuring 25 mm × 25 mm × 1.5 mm for sol–gel experiments and 100 mm × 25 mm × 1.5 mm for PE-CVD experiments. All the samples were cleaned in an ultrasonic bath with distilled water for 20 min before and after laser texturing.

### 2.2. Laser Texturing

Laser texturing was performed with a LaserPoint YFL 20P nanosecond pulsed fiber laser with emission wavelength of 1064 nm, maximum average power of 17 W, and repetition rate of 20 kHz. Beam movement was achieved with a galvanometric scanning head and an f-theta lens characterized by a focal length of 160 mm, achieving a focused laser spot diameter of 60 μm.

Five different scanning strategies were employed, including separate and adjacent single pulses, parallel lines, crossed lines, and concentric circles. For each scanning strategy, four tests were carried out over separate 8 mm × 8 mm areas while varying the average laser power from 2–17 W. Separate identical textured areas were prepared for samples to be coated with sol–gel and PE-CVD processes. All laser processing parameters employed during the study are shown in Table 1, while Figure 2 shows schematics of the different scanning strategies employed.

**Table 1.** Summary of laser processing parameters employed for experiments.

| Name | No. | Average Laser Power (P) [W] | Pulse Energy (Ep) [μJ] | Peak Laser Pulse Fluence (F) [J/cm$^2$] | Scanning Speed (v) [mm/s] | Hatch Distance (d) [μm] | Total Energy Dose (Et) [J/cm$^2$] |
|---|---|---|---|---|---|---|---|
| Single Pulses (Separate) | 1 | 2 | 100 | 7.1 | 2400 | | 0.7 |
| | 2 | 7 | 350 | 24.8 | 2400 | 120 | 2.4 |
| | 3 | 12 | 600 | 42.4 | 2400 | | 4.2 |
| | 4 | 17 | 850 | 60.1 | 2400 | | 5.9 |
| Parallel Lines | 1 | 2 | 100 | 7.1 | 300 | | 5.6 |
| | 2 | 7 | 350 | 24.8 | 300 | 120 | 19.4 |
| | 3 | 12 | 600 | 42.4 | 300 | | 33.3 |
| | 4 | 17 | 850 | 60.1 | 300 | | 47.2 |
| Crossed Lines | 1 | 2 | 100 | 7.1 | 300 | | 11.1 |
| | 2 | 7 | 350 | 24.8 | 300 | 120 | 38.9 |
| | 3 | 12 | 600 | 42.4 | 300 | | 66.7 |
| | 4 | 17 | 850 | 60.1 | 300 | | 94.4 |
| Single Pulses (Adjacent) | 1 | 2 | 100 | 7.1 | 1200 | | 2.8 |
| | 2 | 7 | 350 | 24.8 | 1200 | 60 | 9.7 |
| | 3 | 12 | 600 | 42.4 | 1200 | | 16.7 |
| | 4 | 17 | 850 | 60.1 | 1200 | | 23.6 |
| Concentric Circles | 1 | 2 | 100 | 7.1 | 300 | | 5.6 |
| | 2 | 7 | 350 | 24.8 | 300 | 120 | 19.4 |
| | 3 | 12 | 600 | 42.4 | 300 | | 0.7 |
| | 4 | 17 | 850 | 60.1 | 300 | | 2.4 |

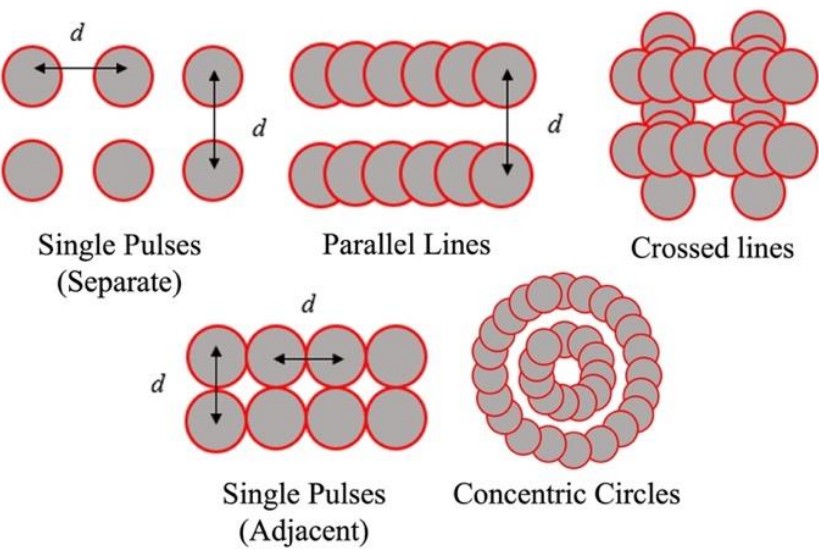

**Figure 2.** Schematic representation of scanning strategies employed for laser texturing experiments (see Table 1).

### 2.3. Sol–Gel Deposition

A first set of samples was coated by employing the sol–gel process, where a colloidal solution (sol) containing monomers was made to form a gel through hydrolysis and condensation reactions. Subsequently, the liquid component was removed via heat treatment, giving rise to nanometric agglomerates. This approach represents a versatile method with which a large range of products can be obtained at a much lower cost than is required for other coating processes such as CVD, evaporation coating, or sputtering. In addition, the sol–gel process allows better control of microstructure and porosity over a larger surface area. Three different sol–gel treatments were performed to determine the effect of the precursor composition and presence of an interlayer on the effectiveness of deposition

on laser-textured substrates. A first series of samples, denominated *Surf 1*, was obtained by carrying out sol–gel deposition using tetraethyl orthosilicate (TEOS) as a precursor. A second series, *Surf 2*, instead employed 3-aminopropyltriethoxysilane (APTES). A third series, *Surf 3*, employed the same precursor as *Surf 2* but with a spin-coated interlayer of polyethylene (PE) to promote adhesion between the substrate and nanoparticles during the process.

The sol–gel treatment was carried out in line with the existing literature [38]. For *Surf 1*, a closed beaker containing ethanol ($C_2H_5OH$), water ($H_2O$), and ammonium hydroxide ($NH_4OH$) with a molar ratio of 1/0.26/0.20, respectively, was stirred for 30 min at a temperature of 40 °C, after which TEOS was added to the solution until a concentration of $4.0 \times 10^{-3}$ mol dm$^{-3}$ was obtained. A support holding the laser-textured samples was then inserted, allowing the magnetic stirrer to rotate beneath the samples as shown schematically in Figure 3. The solution was stirred at a speed of 400 rpm and a temperature of 40 °C for 18 h. At the end of the operation, the samples were extracted from the solution, washed with ethanol, and placed in an oven at 100 °C for 22 h. This final operation was necessary to eliminate solvent and organic residues from the surface, densifying the film and thus improving its structural stability and mechanical properties.

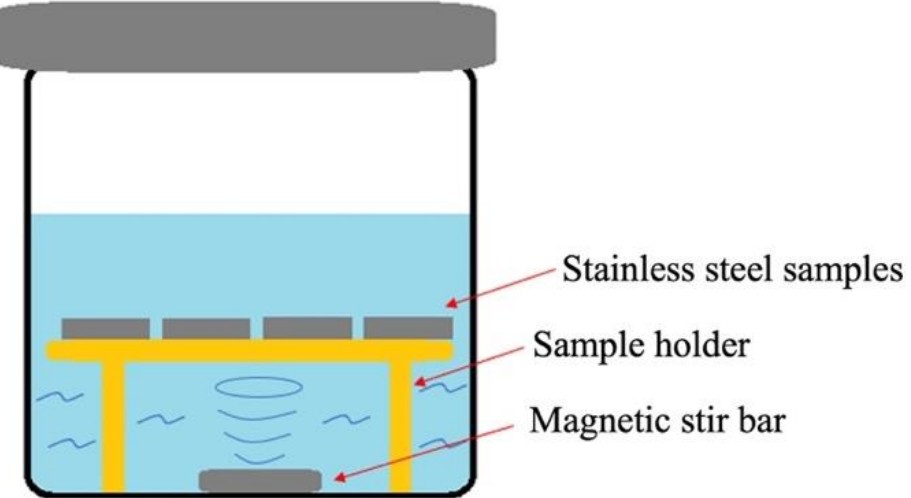

**Figure 3.** Schematic representation of experimental setup employed for sol–gel process.

The same procedure was carried out for *Surf 2* samples but using 3-aminopropyltriethoxysilane (APTES) as a precursor. *Surf 3* samples were prepared with the same method as *Surf 2* but with an intermediate spin-coated polymeric film on the laser-textured substrates. The necessity for such an interlayer was motivated by the requirement to improve adhesion between the sol–gel coating and the underlying metallic surface. The adhesion of silica nanoparticles to polymeric surfaces is well-documented in the literature [39,40]. The molecular chains on the surface of the PE are in fact a good foothold for the nucleation and growth of silica nanoparticles, improving adhesion of the coating to the underlying metallic substrate. Spin coating consisted of depositing 0.0284 g of polyethylene (PE) dissolved in 30 mL of dichloromethane on the surface, as shown in Figure 4, with the latter placed in rotation to spread the solution over the surface by centrifugal force, thinning the liquid through both the spreading and evaporation of the solvent. The successful deposition of thin PE films on laser-textured surfaces was verified via infrared spectroscopy, which highlighted the presence of characteristic peaks directly attributable to the typical polymeric structure of PE.

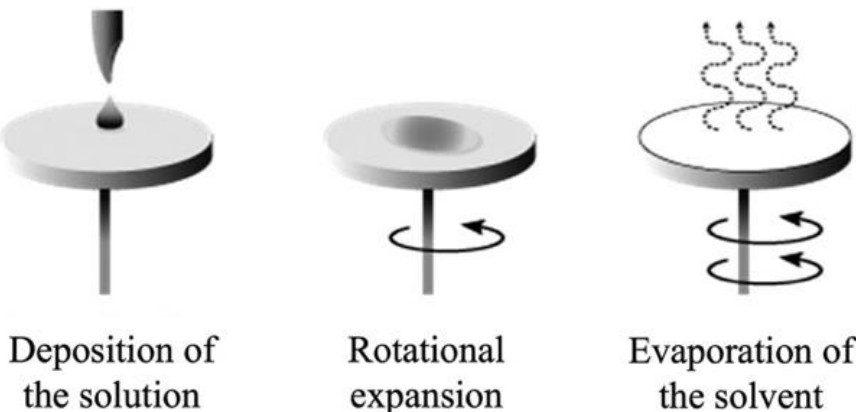

Deposition of the solution    Rotational expansion    Evaporation of the solvent

**Figure 4.** Schematic representation of spin coating process.

### 2.4. PE-CVD Deposition

A second set of samples was coated using PE-CVD deposition, where specific gases were ionized and dissociated through plasma energy, reacting to form a thin film. The CVD technique has progressed enormously over recent years and is now considered one of the most reliable and effective ways of coating materials. It can be applied to a wide range of different materials, including plastics, metals, and ceramics, with negligible environmental impact and virtually no health risk. The main advantage of this technology is the possibility of obtaining a homogeneous and pure film on the substrate. PE-CVD deposition was carried out using a Duralar Centurion, shown in Figure 5. Samples were rotated in a planetary motion while a potential difference was created between the samples and the chamber to optimize treatment uniformity.

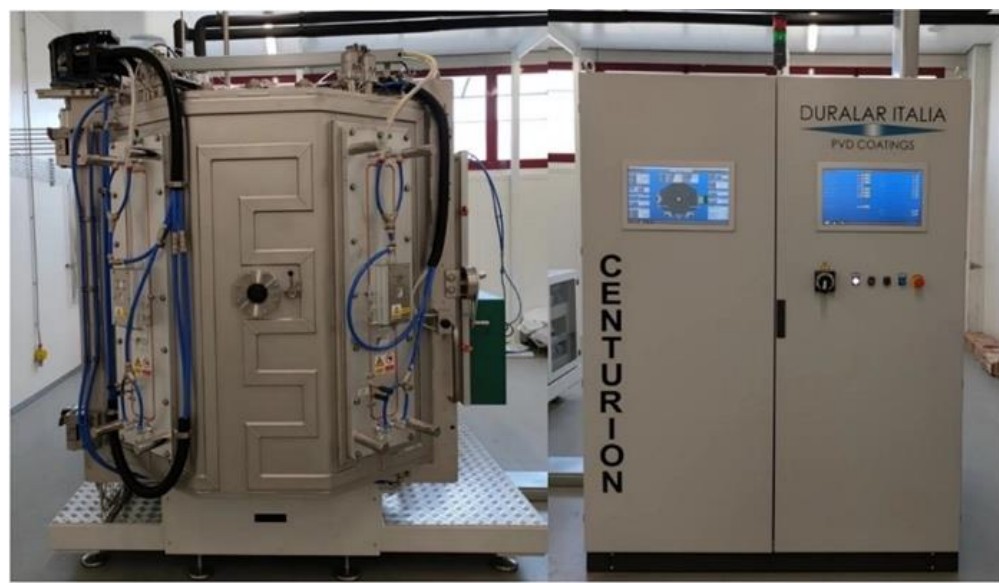

**Figure 5.** Photograph of Duralar Centurion system employed for PE-CVD deposition.

A single series of samples, denominated *Surf 4*, was prepared using hexamethyldisiloxane ($C_6H_{18}OSi_2$) as a precursor for plasma deposition. During the process, helium was used as a carrier gas to transport the precursor inside the chamber. An initial vacuum and heating stage was performed to achieve steady-state conditions within the chamber. A subsequent glow stage was performed to activate and clean the substrates, with Argon (Ar) gas made to flow into the chamber while a pulsed electric field (bias) was applied between the frames and walls of the chamber. Plasma in the glow discharge regime developed around the substrates, with consequent $Ar^+$ ion bombardment exploited to remove

contaminants from the surface. Finally, deposition was performed to deposit a thin film of $SiO_x$ onto the samples. The parameters used for plasma deposition are provided in Table 2.

**Table 2.** Summary of PE-CVD deposition parameters employed for coating of *Surf 4* samples ($SiO_x$ 30 g/h, 400 sccm, $O_2$ RF 3 kW).

| Phase | Duration (s) | T (°C) | P (Pa) | Process Gas | | Gas Carrier | | Sources | |
|---|---|---|---|---|---|---|---|---|---|
| | | | | Type | Scope (sccm) | Type | Scope (sccm) | Type | Electrical Parameters |
| Vacuum and heating | 900 | 150 | 0.05 | | | | | | |
| Glowing | 600 | 120 | 1.4 | Ar | 1500 | | | Bias | 650 V 1.7 A |
| SiOx deposition | 1000 | 120 | 1 | O₂ | 400 | He | 250 | Bias | 50 V 0.9 A 100 kHz 1 µs |
| | | | | | | | | RF | 3 kW |

### 2.5. Surface Characterization

Coherence scanning interferometry (CSI) was used to acquire the surface topography of samples after laser texturing and again after sol–gel and PE-CVD deposition. A Taylor Hobson CCI MP-I optical profilometer was employed, equipped with a 50× objective characterized by a numerical aperture of 0.55 and a working distance of 3.4 mm. With this setup, a vertical resolution of <1 nm and a horizontal resolution of <1 µm was achieved over an area of 336 µm × 336 µm, corresponding to 1024 × 1024 pixels. Data acquisition and processing was performed with TalyMap™ software to obtain digital topographic maps, section profiles, and areal roughness parameters. $S_z$ was determined as the peak-to-valley height over the acquired area, while the arithmetical mean height, $S_a$, skewness, $S_{sk}$, and kurtosis, $S_{ku}$, were calculated in line with ISO 25178:

$$S_a = \frac{1}{N_x N_y} \sum_{i=1}^{N_x} \sum_{j=1}^{N_y} |z_{i,j}| \tag{1}$$

$$S_{sk} = \frac{1}{S_q^3} \frac{1}{N_x N_y} \sum_{i=1}^{N_x} \sum_{j=1}^{N_y} z_{i,j}^3 \tag{2}$$

$$S_{ku} = \frac{1}{S_q^4} \frac{1}{N_x N_y} \sum_{i=1}^{N_x} \sum_{j=1}^{N_y} z_{i,j}^4 \tag{3}$$

where $z_{i,j}$ is the height of point $(i,j)$ with respect to the mean plane, $N_x$ and $N_y$ are the number of equally-spaced data points in the $x$ and $y$ directions, and $S_q$ is the root mean square height:

$$S_q = \sqrt{\frac{1}{N_x N_y} \sum_{i=1}^{N_x} \sum_{j=1}^{N_y} z_{i,j}^2} \tag{4}$$

Scanning electron microscopy (SEM) and electron dispersive X-ray spectroscopy (EDX) were used to perform detailed analysis of the surface morphology and chemical composition, respectively. A focused ion beam (FIB) with a Gallium source was used to ablate small areas of the coated surfaces to allow cross-sectional analysis of the surface layers to be undertaken. SEM, EDX, and FIB were performed with a Zeiss Auriga Compact SEM-FIB. The instrument was used in cross-beam mode, with the sample positioned at the intersection between the electron and ion beams to allow the area ablated by the ion beam to be visualized by SEM imaging. This setup allowed precise identification of the ablated

area and its evolution over time. Incisions were made via FIB on *Surf 3* and *Surf 4* samples. A trapezoidal incision was chosen to maximize the viewing angle of the cross-section, allowing the morphology and thickness of the deposited films to be evaluated.

## 3. Results and Discussion

### 3.1. Morphology and Topography of Laser-Textured Surfaces

Figure 6 shows 3D topography maps of the five textures made by laser texturing at an average power of 17 W. Laser texturing created grooves and craters by laser ablation, where absorption of laser energy resulted in a rapid increase in surface temperature leading to the vaporization and ejection of molten material from the region exposed to the laser beam. Further to material removal, laser ablation in the nanosecond regime also led to melting and solidification of material around the edges of grooves, generating irregular bumps and ridges that increased the surface roughness. Strategies employing a laser scanning speed of 300 mm/s (parallel lines, crossed lines, and concentric circles) led to the creation of grooves in correspondence with regions exposed to the laser beam due to the relatively high pulse overlap, 75% of the focused laser spot diameter (60 µm). Strategies employing high scanning speed (separate and adjacent single pulses) instead led to the generation of individual ablation craters resulting from material ejection during each laser pulse with no overlap.

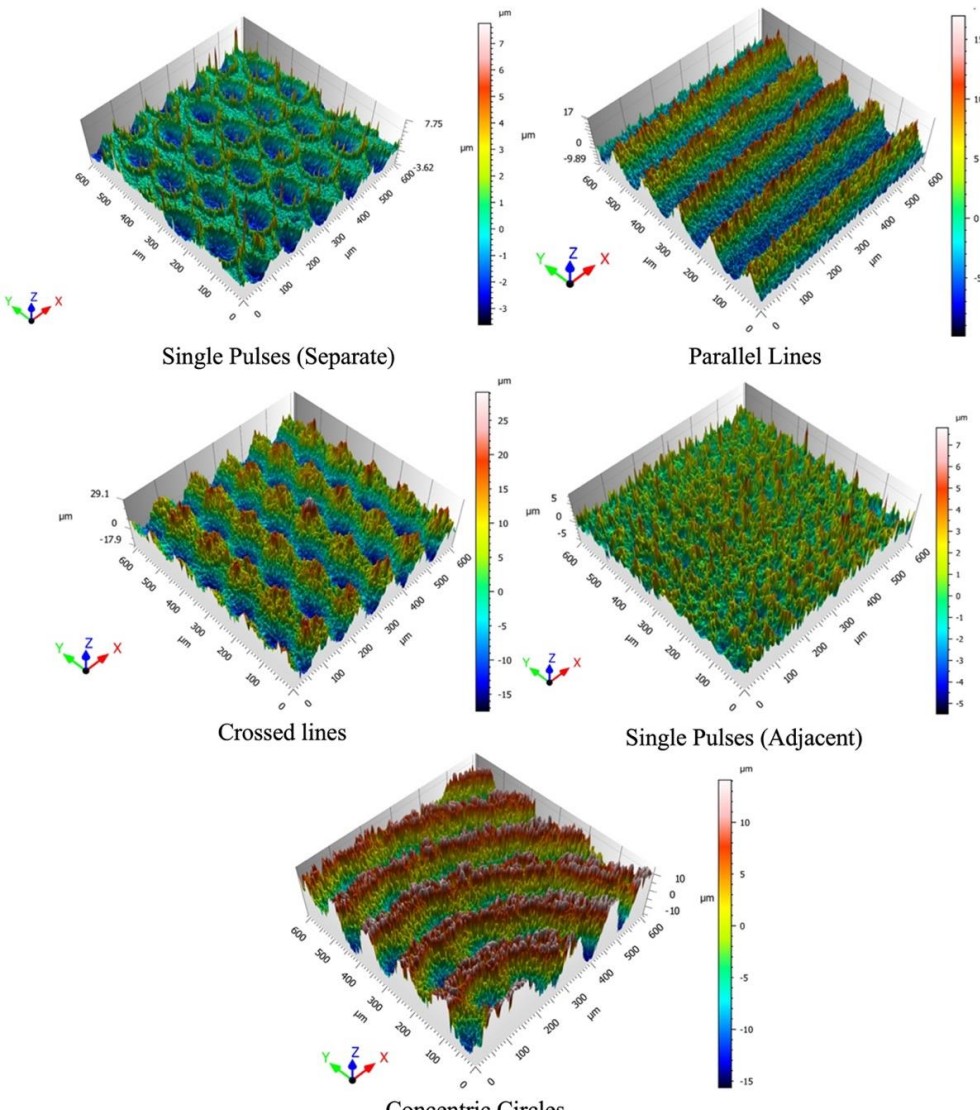

**Figure 6.** Three-dimensional topography maps of laser-textured surfaces without coating obtained with all scanning strategies and an average laser power of 17 W.

An example of structures achieved at lower laser power can be seen in Figure 7, which presents SEM images of samples textured with the parallel lines strategy at 2 W and 7 W. It can be seen that both the ablation depth and the amount of remelted material increased with laser power owing to the larger quantity of energy deposited. The surface topography was not significantly affected by remelted material outside the exposed region at the lowest tested laser power (Figure 7a), with grains of the original substrate material visible in regions immediately adjacent to the grooves. At higher power (Figure 7b), a larger ablated region was evident, while material remelt led to irregular bumps and ridges both within each groove and in the adjacent area.

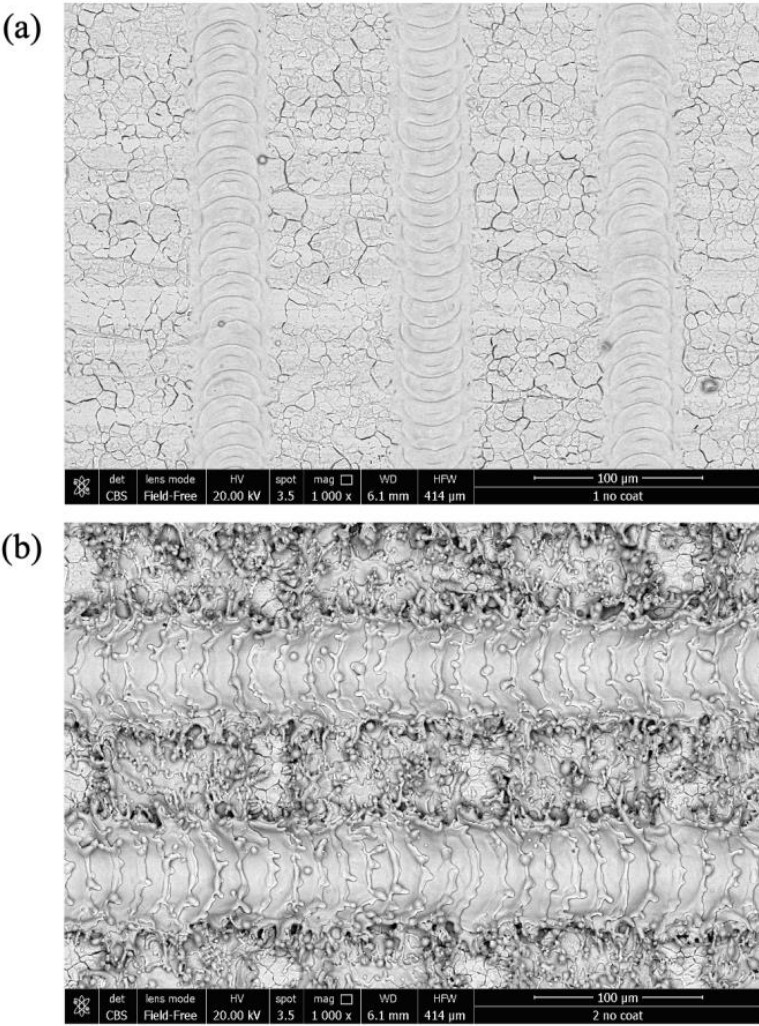

**Figure 7.** SEM images of laser-textured surfaces without coating obtained with parallel lines scanning strategy: (**a**) 2 W; (**b**) 7 W.

Figure 8 compares measured values of $S_a$, $S_z$, $S_{sk}$, and $S_{ku}$ for all the scanning strategies and laser powers. Increasing the laser power led to deeper craters and grooves, which is reflected in the measured values of both $S_a$ and $S_z$. Separate and adjacent single pulses were nonetheless limited to maximum values of approximately $S_a = 1$ μm and $S_z = 14$ μm, while strategies employing a scanning speed of 300 mm/s achieved much higher surface roughness, up to $S_a = 6.4$ μm and $S_z = 47$ μm for the crossed lines strategy with an average laser power of 17 W. $S_{sk}$ was between $-1.5$ and $-0.5$ for all strategies at 2 W owing to the presence of smaller, separate ablation craters or grooves. This led to a generally flat surface characterized by discrete, separate valleys. $S_{sk}$ then increased with increasing laser power to approximately 0 for single pulses, crossed lines, and concentric circles, and 0.5–1 for

adjacent single pulses and parallel lines. These outcomes imply that the height distribution was predominantly symmetrical, with slightly positive skewness in some cases due to distinct peaks resulting from the presence of remelt adjacent to craters and grooves, which can be seen in Figure 6. $S_{ku}$ was in the range 4.5–6.5 for all strategies at 2 W, implying a slightly leptokurtic height distribution characterized by sharp or indented portions. This parameter then steadily decreased with increasing laser power to 2–4 for all scanning strategies, implying that the height distribution was predominantly mesokurtic at high laser power.

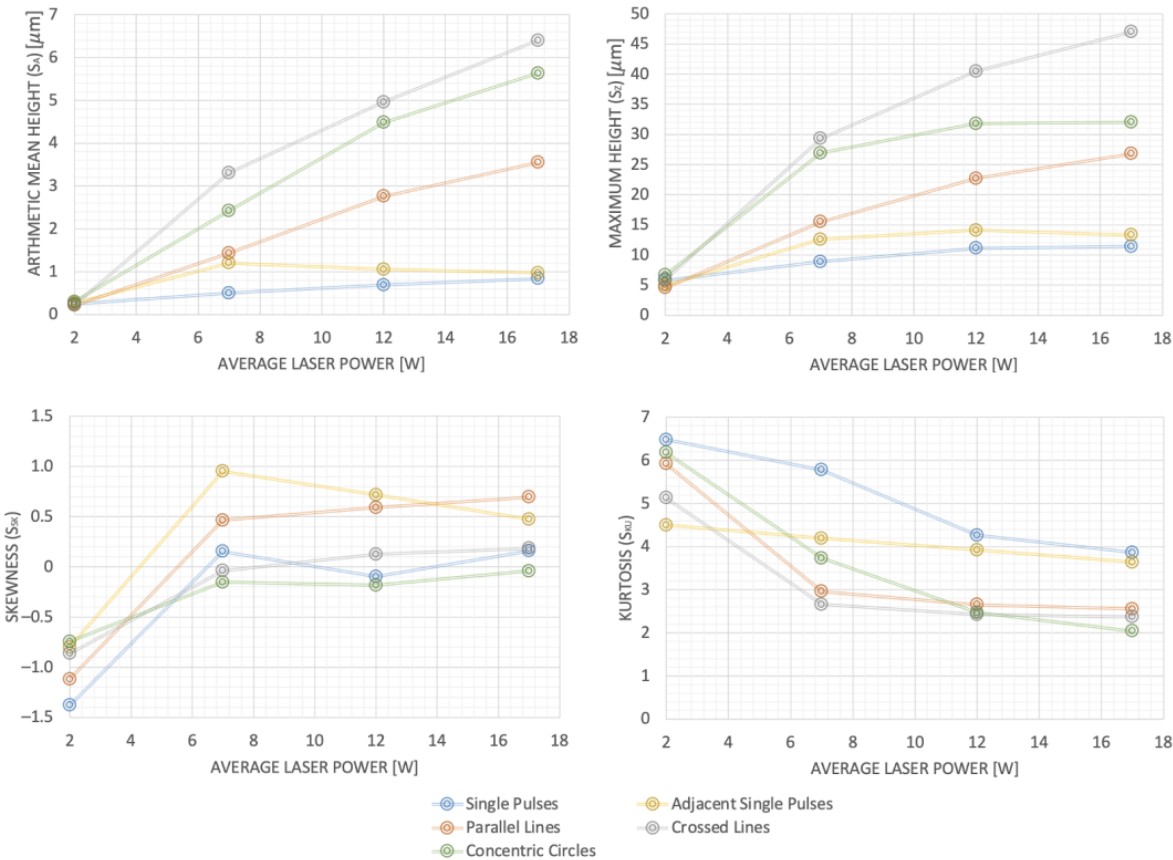

**Figure 8.** Arithmetic mean height, $S_a$, maximum height, $S_z$, skewness, $S_{sk}$, and kurtosis, $S_{ku}$, of laser-textured surfaces without coating as a function scanning strategy and laser power.

### 3.2. Morphology and Topography of Coated Surfaces

Figures 9 and 10 show 3D topography maps of *Surf 1* and *Surf 4*, where laser texturing was combined with sol–gel (TEOS) and PE-CVD coatings, respectively. In general, deposition of the coatings led to no qualitative difference when compared to surfaces subject to laser texturing alone (Figure 6). It must be noted that the profile corresponding to *Surf 1* concentric circles in Figure 9 (laser texturing + sol–gel (TEOS)) was acquired further from the center of the concentric circles than in the previous case, leading to lower groove curvature within the measured region. This surface was nonetheless qualitatively similar to the uncoated samples over the entire textured area. The obtained outcomes suggest that the surface topography obtained with laser ablation remains similar after sol–gel and PE-CVD deposition, implying that the proposed hybrid approach is feasible in terms of achieving coated surfaces with well-defined surface topography.

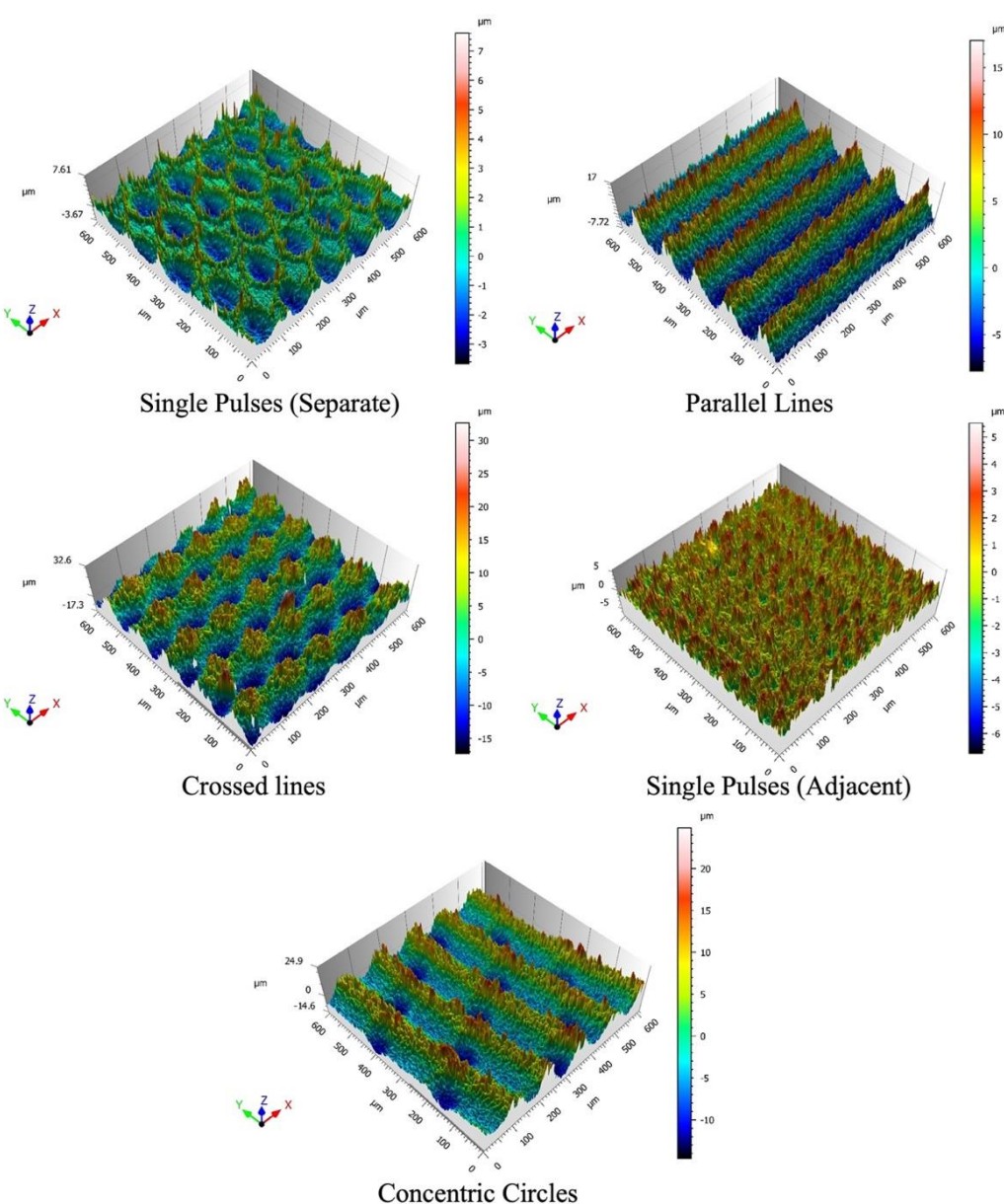

**Figure 9.** Three-dimensional topography maps of *Surf 1* samples (laser texturing + sol–gel (TEOS)) for all scanning strategies and an average laser power of 17 W.

To confirm this outcome, $S_a$, $S_z$, $S_{sk}$, and $S_{ku}$ were quantified for *Surf 1*, *Surf 3*, and *Surf 4*, all subject to laser texturing with the parallel lines scanning strategy and different coating procedures (Figure 11). As power increased, so too did $S_a$ and $S_z$ in all cases, with more intense laser ablation leading to larger craters and ridges. $S_{sk}$ increased from approximately −1 at 2 W to approximately 0.5 over the same range, while $S_{ku}$ decreased from 4.5–6 to approximately 2.5. Sol–gel coating led to an insignificant difference in roughness compared to surfaces subject to laser texturing alone; however, introduction of the PE interlayer led to an increase in $S_a$ of approximately 0.5 μm and $S_z$ of approximately 6 μm at the highest tested laser power. Minor differences in $S_{sk}$ and $S_{ku}$ were also observed at low laser power; however, these were no longer significant at all other tested power levels. PE-CVD led to insignificant changes in surface roughness parameters compared to laser texturing alone. While introduction of the PE interlayer prior to sol–gel deposition clearly led to measurable effects on the resulting topography, differences in roughness parameters were nonetheless limited. Sol–gel and PE-CVD alone, on the other hand, led to accurate transfer of the underlying surface topography obtained by laser ablation. This outcome confirms that the

approach allows surface topography to be tailored independently via laser ablation, with subsequent sol–gel or PE-CVD coatings having negligible effect on the resulting arithmetic mean surface roughness.

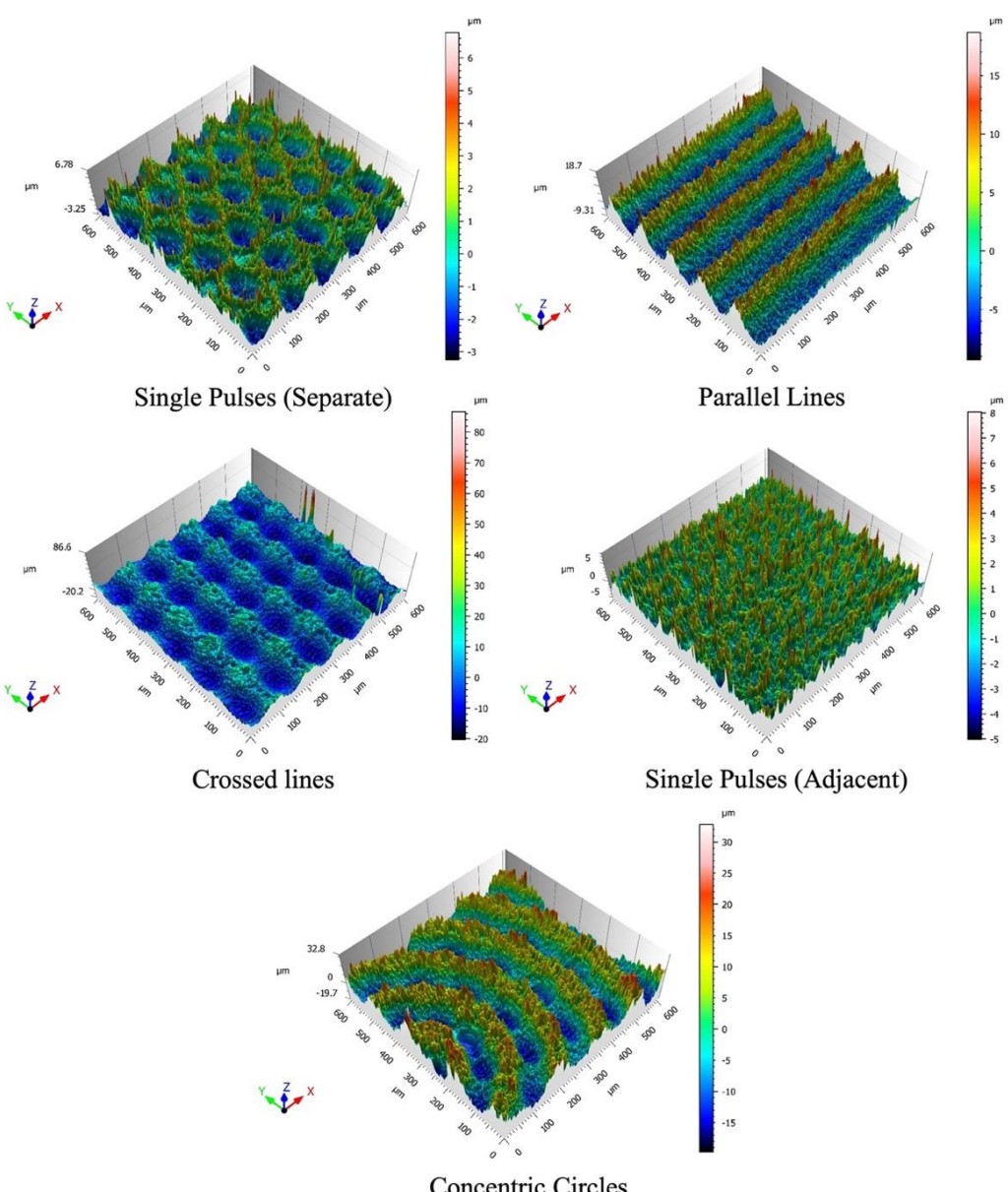

**Figure 10.** Three-dimensional topography maps of *Surf 4* samples (laser texturing + PE-CVD) for all scanning strategies and an average laser power of 17 W.

Figure 12 displays SEM images of surfaces obtained with the parallel lines strategy at 7 W without coating (Figure 12a) and after sol–gel coating with TEOS as a precursor (*Surf 1*, Figure 12b). At a morphological level there is no visible difference. The lack of nanoparticles on the coated surface may have been due to lack of adhesion during the coating process or deposition in the form of a homogeneous layer without the formation of nanoparticles, indistinguishable from the laser-textured surface. Figures 13 and 14 show SEM images of *Surf 3* and *Surf 4* samples, respectively, obtained with the parallel lines strategy at an average laser power of 12 W. At 2,000× and 10,000× magnification, it is possible to observe the morphology created by laser treatment within the exposed area, characterized by ripples and protrusions generated by the ablation phenomenon. At 40,000× magnification, nanoparticles associated with the coatings can be observed in both

cases. It can be seen that sol–gel deposition with a PE interlay and PE-CVD deposition took place in a homogeneous and compact manner within features generated by laser ablation. The roughness generated by laser texturing therefore promoted good adhesion of the coating in both cases. Figures 15 and 16 present SEM cross-sections of incisions obtained by FIB machining of the same surfaces. In both cases, it can be observed that the sol–gel and PE-CVD coatings were in the order of 200–400 nm thick with the employed parameters. The coatings were consolidated and compact, confirming complete coverage of the underlying laser-textured substrate. The effective coating of laser-textured metallic surfaces is therefore feasible with both sol–gel and PE-CVD processes.

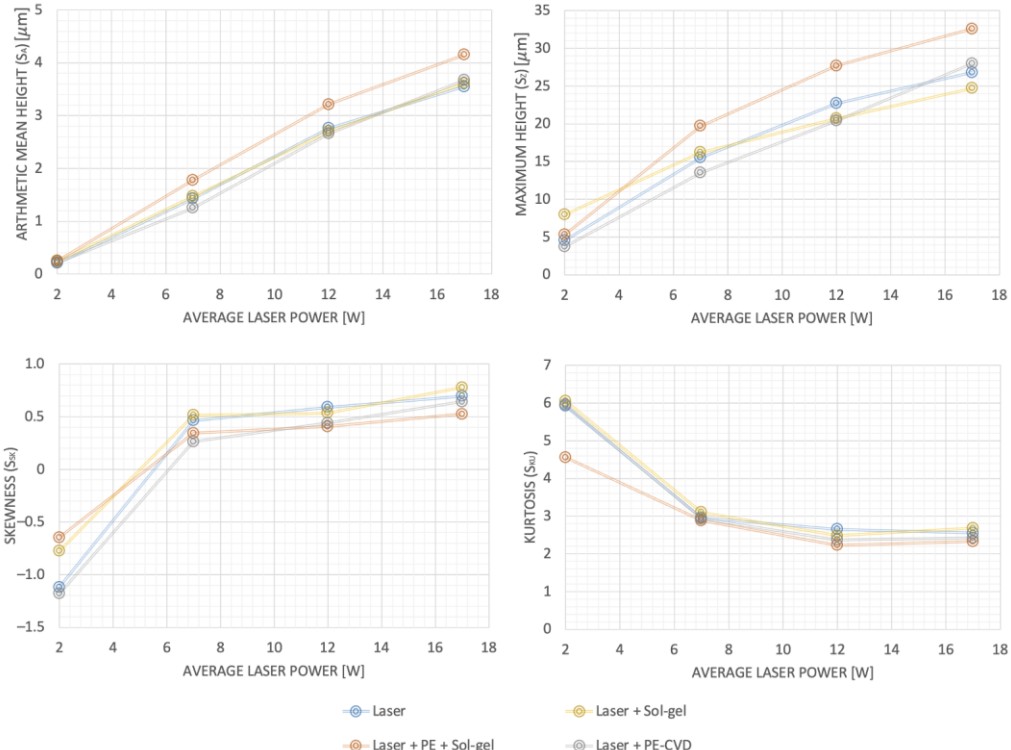

**Figure 11.** Comparison of arithmetic mean height, $S_a$, maximum height, $S_z$, skewness, $S_{sk}$, and kurtosis, $S_{ku}$, of laser-textured surfaces without coating and *Surf 1* (laser texturing + sol–gel (TEOS)), *Surf 3* (laser texturing + sol–gel (APTES + PE interlayer)), and *Surf 4* (laser texturing + PE-CVD) samples obtained with parallel lines scanning strategy.

### 3.3. Surface Chemistry

The results of SEM-EDX analyses performed on the same surfaces are shown in Figures 17 and 18. EDX spectra confirmed the presence of iron (Fe), chromium (Cr), and nickel (Ni), which can be attributed to the metallic substrate exposed during FIB machining, as well as silicon (Si), which can be attributed to the applied coatings. Comparing the spectra for *Surf 3* and *Surf 4*, the presence of Si is much higher in the latter case, suggesting that PE-CVD exhibited a more pronounced effect on the surface chemical composition than the sol–gel treatment with a PE interlayer. Further investigation into this aspect is now required. The color maps of Si provide a qualitative representation of the distribution of this element on the sample surface. A distribution of particles containing silicon can be observed on the portion of the surface under examination, while the underlying substrate exhibits much lower quantities. The presence of Si below the coatings can be attributed to residual Si within the material itself (<1% for AISI 304), as well as the deposition of ablated material during FIB machining. The outcomes of SEM-EDX analysis nonetheless confirm the effective modification of surface chemistry for both sol–gel and PE-CVD treatments, in line with the presence of silica nanoparticles.

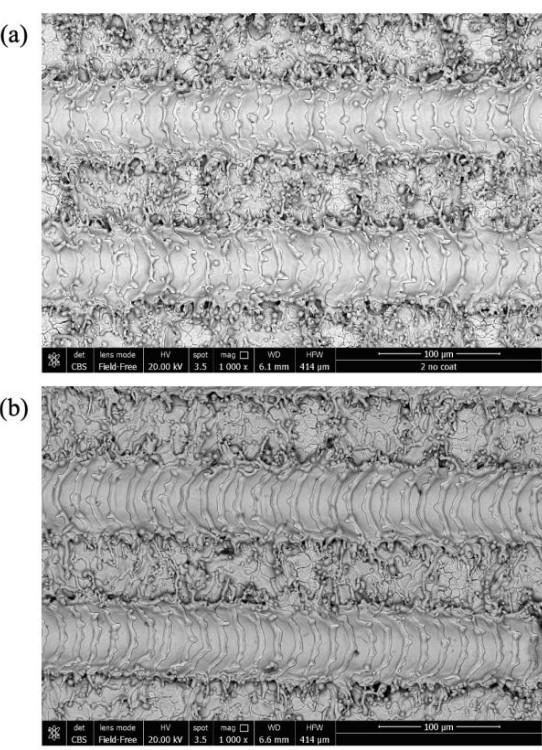

**Figure 12.** SEM images of surfaces obtained with parallel lines scanning strategy at 7 W: (**a**) laser texturing only; (**b**) *Surf 1* sample (laser texturing + sol–gel (TEOS)).

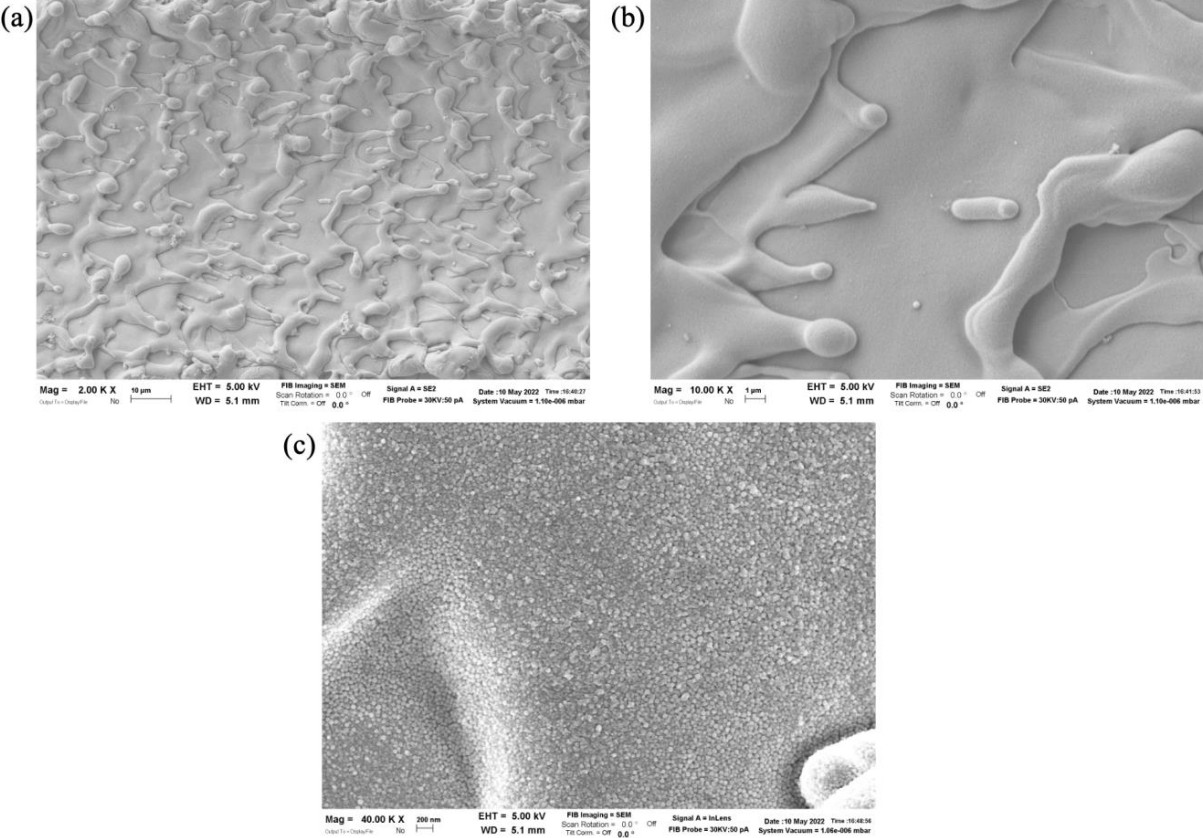

**Figure 13.** SEM images at (**a**) 2000×, (**b**) 10,000×, and (**c**) 40,000× of surfaces obtained with parallel lines scanning strategy at 12 W: *Surf 3* sample (laser texturing + sol–gel (APTES + PE interlayer)).

**Figure 14.** SEM images at (**a**) 2000×, (**b**) 10,000×, and (**c**) 40,000× of surfaces obtained with parallel lines scanning strategy at 12 W: *Surf 4* sample (laser texturing + PE-CVD).

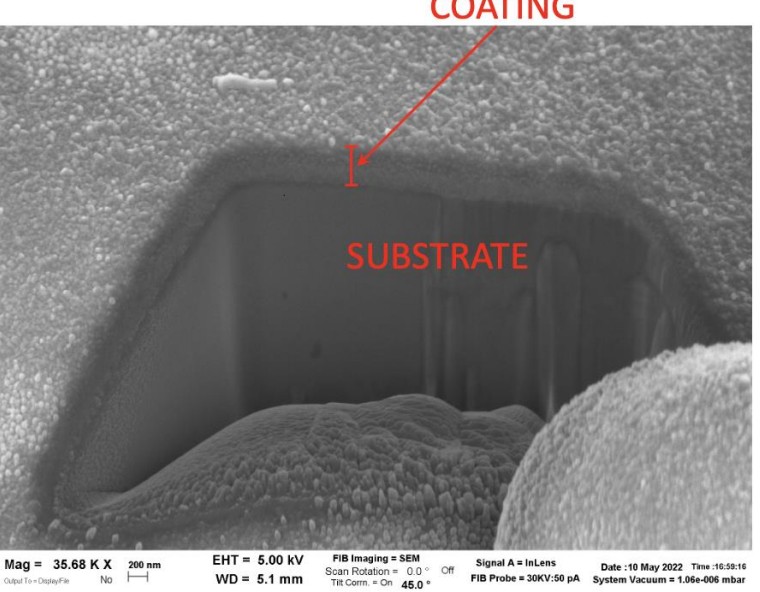

**Figure 15.** SEM image of cross-section obtained by FIB machining of surface textured with parallel lines scanning strategy at 12 W: *Surf 3* sample (laser texturing + sol–gel (APTES + PE interlayer)).

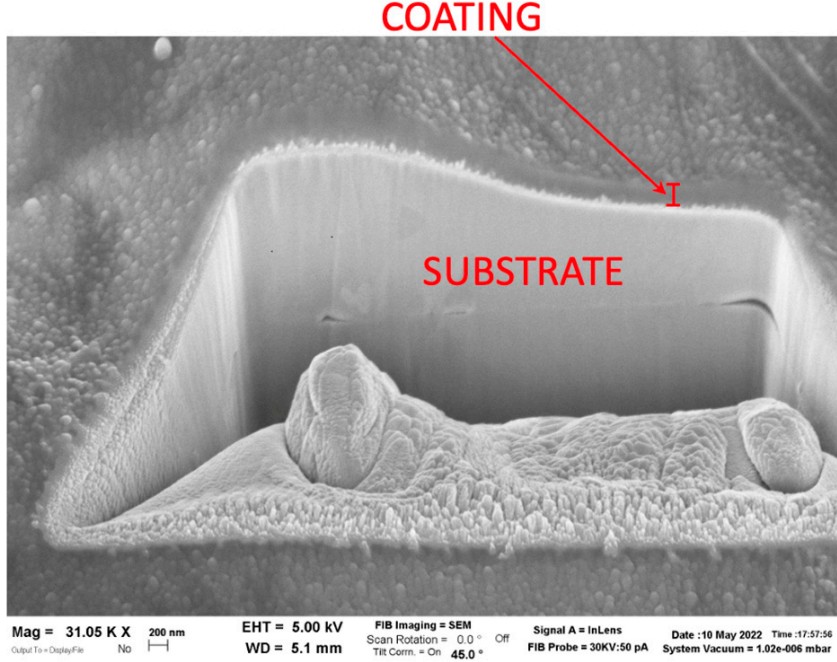

**Figure 16.** SEM image of cross-section obtained by FIB machining of surface textured with parallel lines scanning strategy at 12 W: *Surf 4* sample (laser texturing + PE-CVD).

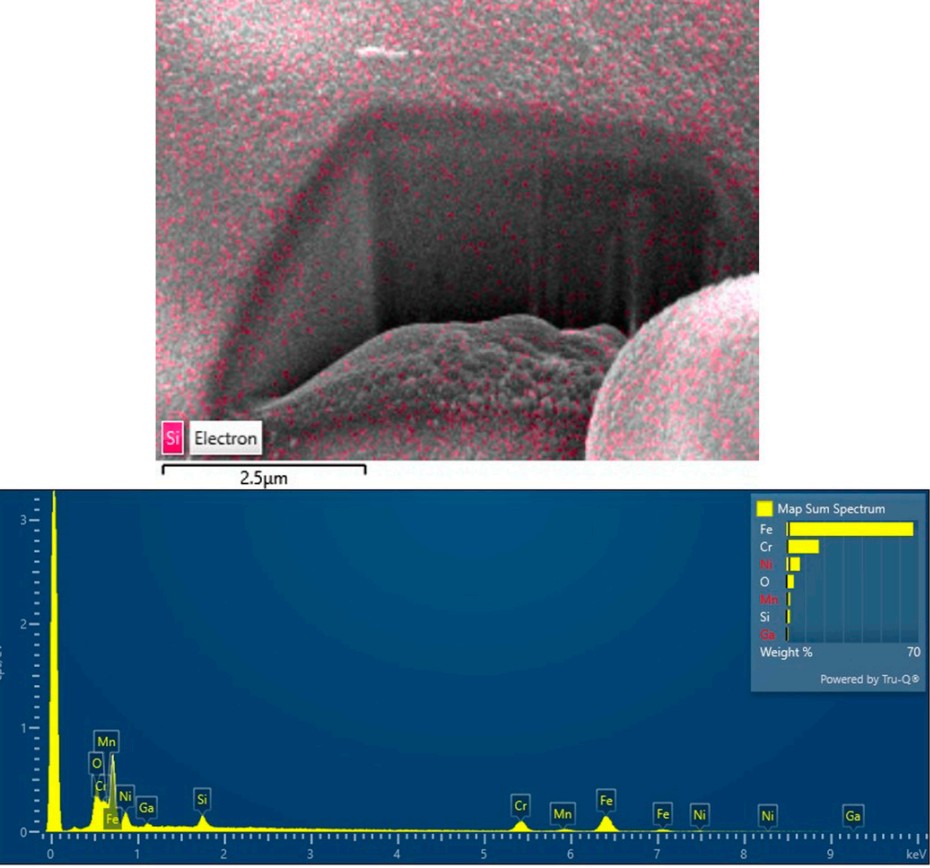

**Figure 17.** EDX spectrum and silicon particle distribution around cross-section obtained by FIB machining of surface textured with parallel lines scanning strategy at 12 W: *Surf 3* sample (laser texturing + sol–gel (APTES + PE interlayer)).

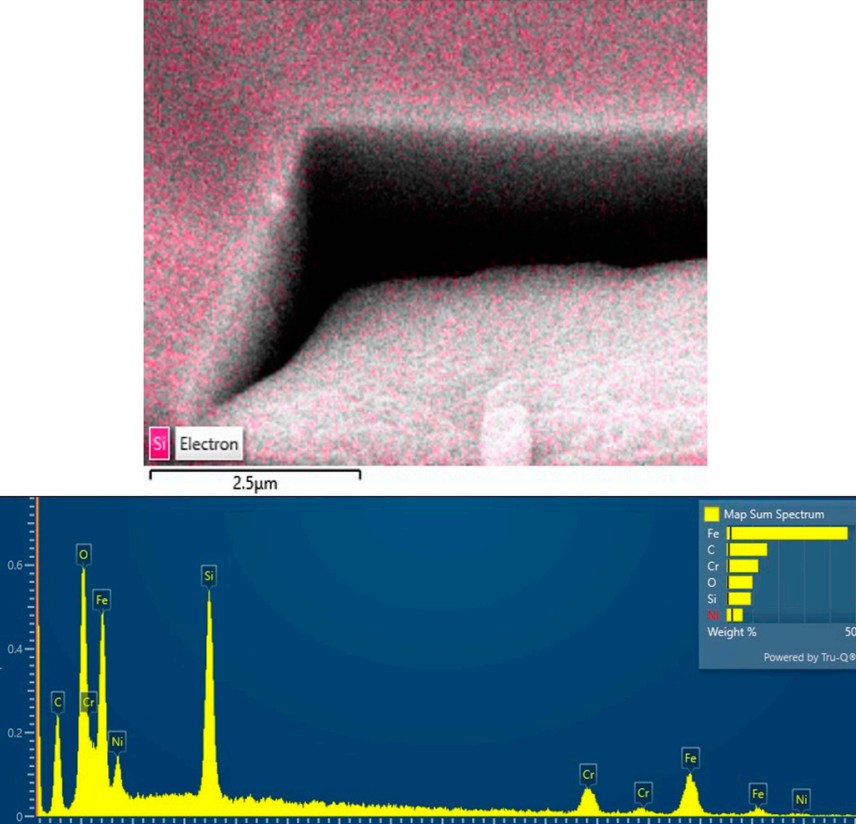

**Figure 18.** EDX spectra and silicon particle distribution around cross-section obtained by FIB machining of surface textured with parallel lines scanning strategy at 12 W: *Surf 4* sample (laser texturing + PE-CVD).

## 4. Conclusions

A hybrid approach for tailoring surface topography and chemistry has been developed based on nanosecond pulsed laser texturing and coating with sol–gel and PE-CVD processes. The aim of this approach was to provide a more effective tool for surface engineering by allowing topography and chemistry to be tailored independently, an outcome that is currently difficult or slow to achieve with existing methods. By carefully selecting process parameters to achieve coatings of less than 1 μm, it was found that the topography of the underlying laser-textured metallic surface could effectively be preserved, with minimal impact on the surface topography and arithmetic mean surface roughness ($S_a$). It was nonetheless observed that coating of laser-textured surfaces required careful selection of process parameters to ensure that adhesion was sufficient to promote deposition, but thickness was limited to avoid negatively impacting the resulting topography. In the case of sol–gel deposition, the introduction of a PE interlayer was found to improve adhesion while increasing the value of $S_a$ by approximately 0.5 μm compared to the laser-textured substrate.

Further investigation is now required into different laser texturing approaches, including smaller features such as submicrometric laser-induced periodic surface structures (LIPSS) and hierarchical surface structures that can be obtained with ultrashort pulsed laser irradiation. Furthermore, the mechanical and chemical durability of coatings deposited on laser-textured substrates must be investigated to ensure adequate longevity of the obtained surfaces. Finally, the resulting functional properties must be assessed for applications of surface engineering based on specific requirements in terms of surface topography and chemistry. With the possibility of selecting materials for sol–gel deposition and PE-CVD processes, the proposed approach holds promise for tailoring wettability in relation to non-polar liquids that require specific combinations of surface energy and roughness.

**Author Contributions:** Conceptualization, A.H.A.L., C.S. and L.R.; methodology, all authors; validation, all authors; writing—original draft preparation, M.B. and A.H.A.L.; writing—review and editing, all authors; supervision, L.R. All authors have read and agreed to the published version of the manuscript.

**Funding:** This research received no external funding.

**Conflicts of Interest:** The authors declare no conflict of interest.

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
