# Peer review of "A Hybrid Approach to Surface Engineering Based on Laser Texturing and Coating"

_jmmp, doi:10.3390/jmmp7020059_

Round 1
Reviewer 1 Report
The authors presented a study on a hybrid approach for surface modification by combining laser surface texturing with subsequent coating application. The proposed method may offer advantages over the various existent surface modification techniques. However, the investigation stays mainly at the initial stage of the investigation. Below are some questions for the authors to address.
1. It seems only one set of conditions is used for the sol-gel deposition and one set for the PE-CVD deposition process. How are these conditions determined? Are these optimal conditions?
2. In Figure 8 3D profiles, the bottom concentric circles seem like parallel lines. Please clarify.
3. A unit should be provided for the arithmetic mean height in Figs. 7 and 10.
4. Only Sa is used as a measure for surface profiles in this study and it shows that coating has little effect on it. How about other surface parameters like peak-to-valley height, skewness, and kurtosis?
4. The coating layer in Figs. 14 and 15 should be labeled to provide clarity, especially in Fig. 15, which is hard to tell.
5. It is stated that the introduction of a PE interplay improves adhesion, but without any measurements. Hence, quantitative assessment of adhesion strength is needed to compare the various coatings deposited on the various laser textured surfaces to provide a clear picture for the effects of the coating method and surface texture on adhesion.
Reviewer 2 Report
1. A schematic diagram may be added to illustrate the proposed hybrid method for surface engineering, including laser texturing and the followed coating process.
2. The unique advantages of the proposed method may be further described, or the novelty of the paper should be stressed.
Round 2
Reviewer 1 Report
The authors have addressed the review comments reasonably well.